# Eye Movements in Response to Pain-Related Feelings in the Presence of Low and High Cognitive Loads

**DOI:** 10.3390/bs10050092

**Published:** 2020-05-20

**Authors:** Ramtin Zargari Marandi, Camilla Ann Fjelsted, Iris Hrustanovic, Rikke Dan Olesen, Parisa Gazerani

**Affiliations:** Department of Health Science and Technology Faculty of Medicine, Aalborg University, 9220 Aalborg East, Denmark; ramtin.zargari@gmail.com (R.Z.M.); camilla_fjelsted@hotmail.com (C.A.F.); iris.hrustanovic@gmail.com (I.H.); rikkeolesen92@gmail.com (R.D.O.)

**Keywords:** pain memory, attention, auditory stimuli, oculomotor system, psychological pain

## Abstract

The affective dimension of pain contributes to pain perception. Cognitive load may influence pain-related feelings. Eye tracking has proven useful for detecting cognitive load effects objectively by using relevant eye movement characteristics. In this study, we investigated whether eye movement characteristics differ in response to pain-related feelings in the presence of low and high cognitive loads. A set of validated, control, and pain-related sounds were applied to provoke pain-related feelings. Twelve healthy young participants (six females) performed a cognitive task at two load levels, once with the control and once with pain-related sounds in a randomized order. During the tasks, eye movements and task performance were recorded. Afterwards, the participants were asked to fill out questionnaires on their pain perception in response to the applied cognitive loads. Our findings indicate that an increased cognitive load was associated with a decreased saccade peak velocity, saccade frequency, and fixation frequency, as well as an increased fixation duration and pupil dilation range. Among the oculometrics, pain-related feelings were reflected only in the pupillary responses to a low cognitive load. The performance and perceived cognitive load decreased and increased, respectively, with the task load level and were not influenced by the pain-related sounds. Pain-related feelings were lower when performing the task compared with when no task was being performed in an independent group of participants. This might be due to the cognitive engagement during the task. This study demonstrated that cognitive processing could moderate the feelings associated with pain perception.

## 1. Introduction

In the processing of cognitive tasks, the human brain may prioritize the working memory to give an efficient response. This, in turn, influences the outcome, which may differ if the cognitive load changes (e.g., from low to high) or if the task is performed with ongoing emotions, such as pain, fear, and anxiety [1,2]. Pain is a subjective feeling with several dimensions, one being the affective dimension [3], which contributes to overall pain perception. Fear, anxiety, and memory of past pain experiences may even precede the sensation of pain [2,4,5]. Many researchers have attempted to identify ways to measure pain in an objective manner, for example, brain imaging and assessing the activity of certain brain regions. The reflection of pain on ocular events has been studied less frequently [6,7], and mostly research has been limited to pupillary responses. Eye tracking can provide data required for detecting ocular events and can, thereby, be used to compute oculometrics, i.e., quantitative indices characterizing eye movements. Eye tracking is superior in terms of its practicality, cost, and unobtrusiveness compared with other techniques, e.g., functional magnetic resonance [5]. Ocular events have specific functions. Saccades, for instance, are fast eye movements that direct the foveal part of the retina (providing accurate vision) to salient areas in the visual field. Subsequently, fixations are required to capture visual information. Furthermore, blinks may occur as short eyelid closing and opening movements to, among other purposes, clean and lubricate the surface of the cornea and conjunctiva. Pupil size may change to direct a certain amount of light into the eyes [8] or in response to auditory emotional stimuli in connection with the autonomic nervous system [9]. Ocular events can be influenced by various psychophysiological factors, including emotions [5,10,11,12]. This introduces the possibility of investigating the affective domain of pain by affective computing, where human–machine interactions involving pain can be tested.

Pain may be associated with fear from an uncertain threat. Painful experiences are memorized to avoid similar experiences in the future [13]. Sensory information that directs attention to painful experiences, such as the sound of dental drilling, may lead to the recall of a pain memory that, by itself, may provoke a centrally driven pain experience [14]. For example, the sounds of dental treatment devices (e.g., an ultrasonic dental scaler and a dental turbine) have been shown to significantly reduce cerebral blood flow in patients who have undergone an unpleasant dental experience [15]. Pain might thus be perceived even without physical injury, for instance, due to empathy for pain [16,17] or, in individuals with high levels of mathematics anxiety, by thinking about math [2]. Of note, pain perception is sometimes merely psychological, suggesting that pain may not necessarily be stimulated by physical injury [18,19]. Furthermore, it has been shown that in some chronic pain conditions, pain can remain, even though the injury that caused the pain has long been healed, due to remaining traces of pain memory [20,21]. These factors underline the role of memory and emotions in pain perception [22,23]. Cognitive and affective aspects of pain, for example, phantom headaches, have thus been subject to speculation in recent years [22].

Unpleasant sounds have previously been used to induce pain-related feelings [13]. Unpleasant sounds may evoke negative emotional responses depending on the background experience of an individual, and, therefore, they are associated with a high subjective level of pain [24,25]. Unpleasant auditory stimuli, e.g., chainsaw noises, have been shown to evoke a negative affective state, regardless of the cultural or ethnic background of an individual [25].

Pain perception is further dependent on bottom-up factors, such as the unpredictability of pain, which captures attention in an automatic fashion [1,26,27,28]. Cognitive tasks concurrently engage people’s attention to allocate cognitive capacity in competing choices [29]. Top-down control entails cognitive control of the attention used to process pain [1,27]. Thus, attention and cognitive capacity may together affect pain perception. Pain processing and cognitive functions may reside in common brain networks [26]. Previous studies provided different implications for the association between attention and pain [25,28]. In the study by Romero and colleagues [28], pain perception was reduced while performing cognitive tasks, highlighting the potential of cognitive tasks to distract an individual from pain. Cognitive demands have also been shown to attenuate pain-related activation in different brain areas [26]. Chronic pain may impair cognitive processing [30,31]; however, the relationship with pain-related feelings is unclear. 

Previous studies have shown that cognitive load could be reflected in ocular events, including saccades and fixations [32,33]. Eye movements and cognitive processing share specific neuronal circuits including the paramedian pontine reticular formation, superior colliculus, basal ganglia, posterior parietal, frontal cortices, thalamus, and cerebellum. The involvement of these extensive neural networks in modulating eye movements may also affect the association with pain and attention [34,35]. The association between pain-related feelings and cognitive load has rarely been explored in eye movements and pupillary responses. Thus, this study aimed to advance our understanding of the associations between cognitive processing and pain-related feelings to shed light on the role of cognitive load in pain distraction [36,37]. It was hypothesized that the perception of pain-related feelings induced by unpleasant auditory stimuli decreases as the level of cognitive load increases. The effect was hypothesized to be reflected in task performance, oculometrics, and subjective ratings of pain perception.

## 2. Materials and Methods

### 2.1. Participants

Twelve healthy university students (six females and six males) aged 22 to 28 years (mean = 24.8, SD = 2.2) participated voluntarily in this study. All participants had self-reported normal or corrected-to-normal vision. This was further examined by an eye acuity test (the Snellen chart) before the experimental session (Section 2.2). The laterality test, confirmed by the Edinburgh Handedness Inventory [38], showed that all participants were right-handed and used a computer mouse with their right hand. All participants were within the normal hearing range and had symmetric hearing based on The Mimi Hearing Test app, version 4.6.1, examined on an iPhone 6.

The participants were instructed to abstain from alcohol for 24 h, and caffeine, nicotine, and other drugs for 12 h prior to the experimental session. The participants slept for between six and nine hours (mean = 7.6, SD = 1.1) the night before the experimental session. Their normal sleeping hours per night ranged from five to eight hours (mean = 7.3 h). This information was obtained to ensure that there was no confounding effect from these factors.

Before starting the experimental session, all participants signed a written consent form. The experiment was conducted in a quiet laboratory with controlled noise, light, and temperature to avoid any potential disturbing factors. The study was conducted in accordance with the Declaration of Helsinki. Application of the eye tracker was approved by the regional ethics committee (Den Videnskabsetiske Komité for Region Nordjylland, case number: N-20160023).

### 2.2. Experimental Procedure

This study consisted of an experimental session divided into training and main blocks, lasting approximately 90 min in total. The training block contained two cognitive tasks with different levels of difficulty, termed low- and high-load levels. At the end of each task, the participants were instructed to fill out two questionnaires: the National Aeronautics and Space Administration Task Load Index (NASA-TLX) [39] and the Short-form McGill Pain Questionnaire (SF-MPQ) [40], provided in Danish. The study setup is depicted in Figure 1. The participants performed the task twice (5 min each) at two levels of difficulty for training and to become familiar with the questionnaires before starting the main block. The main block consisted of four tasks with low and high levels of cognitive load, once with and once without pain-related sounds. The order of the tasks was randomized across participants. Similar to the training block, each task in the main block was followed by the NASA-TLX and SF-MPQ questionnaires. The NASA-TLX consists of six subscales: frustration, effort, performance, temporal demand, mental demand, and physical demand. The participants rated these six workload measures individually on a continuous scale, ranging from 0 to 100. Subsequently, the workload measures underwent pairwise comparisons where the participants were instructed to choose the workload subscale that corresponded more to the perceived workload, resulting in weighted NASA-TLX scores.

The tasks were performed while sitting on an office chair with the desk surface adjusted to elbow height. The participants were seated approximately 62 cm in front of a 19″ (1280 × 1024 pixels, refresh rate 120 Hz) computer screen. The sounds were played by two Creative Inspire T10 loudspeakers, which were placed at a 25° angle facing the participant and were distanced approximately 65 cm from the right and left ears. The participant’s head was placed on a chin rest, and the computer screen was adjusted to the ear-eye line height for each participant when looking straight at the screen, as shown in Figure 1a.

### 2.3. Auditory Stimuli

A validation study was previously conducted on 14 participants to rate a set of sounds that could be associated with pain-related feelings. The participants who participated in the validation study were different from the participants who participated in the main study (Figure 1) to avoid habituation to the sounds. The participants in the validation study were also volunteers from the same age group as those used for the main study (both university students). For the validation study, 18 different sounds were selected from the database www.soundbible.com, of which six sounds acted as control sounds (e.g., crowd talking, wind blowing, pouring water) and 12 acted as potentially unpleasant sounds that may induce pain-related feelings (e.g., dental scaler, dental turbine, horn, and sawing machine). All sounds were trimmed to five seconds in length and were controlled by the app “Decibel X” (Decibel X: Noise Meter, version 6.2.3) on an iPhone 6 to play in a volume range of 55–60 dB, staying in a safe hearing range. The sounds were played in a randomized order. The participants were asked to fill out a questionnaire for each presented sound. The questionnaire consisted of a modified version of the SF-MPQ and the “Valence” domain of the Self-Assessment Manikin (SAM) questionnaire [41].

The SF-MPQ was used to find the six most unpleasant sounds that could induce pain-related feelings out of the 12 available sounds (Table 1), based on the participants’ answers, which were filled out with paper and pen. This questionnaire is a reliable method for the evaluation and identification of the affective aspects of pain. The SF-MPQ consisted of 15 words, including four descriptors for the affective dimension of pain, as follows: tiring-exhausting, fearful, and punishing-cruel [42]. The participants were instructed to choose a maximum of two of the 15 words to describe how they felt after hearing each sound once. The intensity of the chosen word was then rated as 0 = “no pain”, 1 = “mild”, 2 = “moderate”, or 3 = “severe” [42]. The participants were also instructed to leave out an answer if no word could describe the feeling or if they would rate the sound as inducing “no pain”. Each sound was individually evaluated by the participants after being presented in the validation study.

The SAM questionnaire was used to determine whether each sound was associated with a negative, neutral, or positive feeling. The participants were asked to mark their feelings in a range from one to nine. The SAM scores for valence were interpreted as negative for [1 4), neutral for [4 6], and positive for (6 9]. Brackets and parentheses indicate that the ranges are inclusive and exclusive, respectively.

A total of above 60% of the participants was required to rate a sound as painful in the SF-MPQ score for it to be considered as a candidate sound to induce pain-related feelings. After the evaluation of the questionnaires, six out of the 18 sounds were chosen to induce pain-related feelings. They were assigned as pain-related sounds if they held the highest average scores of the SF-MPQ and were rated as negative in the SAM questionnaire. Additionally, six emotionally neutral sounds based on the average SAM scores and chosen as “no pain” stimuli in the SF-MPQ questionnaire were used as control sounds.

### 2.4. Experimental Task

An interactive computer task (WAME 1.0) [43] was used to impose four conditions: low cognitive load with and without pain-related sounds (respectively denoted by LP and L) and high cognitive load with and without pain-related sounds (respectively denoted by HP and H). MATLAB 2015b (MathWorks, version 8.6.0.267246, MA, USA) was used to design and implement the tasks based on a standard model of computer work [32].

Each cognitive task took 5 min, consisting of 36 and 26 cycles, respectively, of the tasks with low and high levels of cognitive load. Each cycle consisted of three consecutive subcycles, i.e., a memorization period, washout period, and replication period (Figure 1). A cycle began by showing a sequence of a few points connected by straight lines making a random pattern. Each point in the pattern was represented by a different shape, e.g., a plus sign, circle, square, triangle, or pentagram. A short block of text determining the starting point of the pattern was also shown during the memorization period, and the participant was required to memorize the starting point and the pattern prior to the beginning of the washout period when the pattern disappeared. The washout period lasted the same duration as the memorization period. To avoid distractions during the washout period, the mouse cursor became invisible and was re-positioned in the center of the screen, while a black cross appeared at the center of the computer screen to be fixated. This was followed by the replication period, wherein the points of the displayed pattern appeared without the connecting lines. The participants were instructed to click on the points in the correct order to replicate the patterns and to avoid clicking elsewhere.

Feedback was visually provided for the participants to determine whether a click was correct. If a click was correct, a connecting line was depicted between each of the consecutive points. A correct click on the first point was determined by doubling the size of the point. If a click was incorrect, then nothing was changed on the replication panel.

The task was designed to impose low and high levels of cognitive load based on the geometric complexity of the pattern and the number of connecting points, with increased irregularity of the patterns at the high load level compared with the low load level [32]. The memorization and replication periods lasted 2.06 s and 4.11 s, respectively, in the tasks with a low cognitive load (i.e., L and LP). The duration of the memorization and replication periods was set to 2.62 s and 6.02 s in the tasks with a high cognitive load (i.e., H and HP). These durations were assigned based on the methods–time measurement (MTM-100) standard [44].

The control sounds were played randomly during the replication periods of the L and H tasks. Likewise, pain sounds were played randomly during the replication periods of the LP and HP conditions. Half of the cycles in each task included auditory stimuli; this was done to avoid a potential acclimatization effect. After each task, the participants were instructed to respond to the NASA-TLX and the SF-MPQ to indicate their individual perceptions of the cognitive load and pain-related feelings, respectively.

## 3. Data Collection and Analysis

A video-based head-mounted eye tracker (EYE-TRAC^®^ 7) was used to record gaze positions and pupil diameters with a sampling frequency of 360 Hz during the cognitive tasks. All data from the eye tracker were collected on a computer with built-in software (EyeTRAC7, version 1.0.6.0, MA, USA) for the eye tracker. Another computer presented the tasks and synchronized with the computer to record eye tracking data. The calibration of the eye tracker was conducted for each participant before starting the main block with a standard nine numbered calibration point screen. The calibration procedure was conducted before each task in the main block.

Ocular events were extracted from gaze data following the velocity threshold algorithm [45]. Ocular data were segmented as blinks when the pupil was not detected for 100–400 ms. Noisy samples were determined when corneal reflection was not detected or the gaze velocity and acceleration exceeded 500 (°/s) and 50,000 (°/s), respectively. The rest of the gaze samples with velocities higher than 30 (°/s) were considered to be saccades. Further, saccades with a duration exceeding the range of 20–200 ms were excluded due to being noisy samples. The rest of the gaze samples were considered to be fixations if their duration was within the range of 40–2500 ms. Adjacent fixations <11 ms apart were merged into one fixation. The pupil diameter signal underwent linear interpolation for the missing samples during blinks. A three-order zero-phase low-pass Butterworth filter was applied to the pupil diameter signal to remove the noise and artifacts occurring prior to or after blinks [46,47,48].

The maximum value of the gaze velocity during saccades provided the saccade peak velocity. The number of occurrences of saccades, fixations, and blinks during each task divided by the duration (s) of the task yielded the computation of the saccade frequency, fixation frequency, and blink frequency. The pupil dilation range was computed as the range of pupil diameters across cycles. The durations of fixations and blinks were computed, respectively, by the number of data samples divided by the sampling frequency for each fixation and blink.

In addition, the overall performance of the task was calculated to reflect the number of correct and incorrect clicks (accuracy in responding) and the speed of response (the time taken to correctly click on pattern points) [32]. The overall performance level ranged from zero to one, where higher values corresponded to a higher level of performance.

### Statistical Analysis

Statistical analysis was performed using IBM SPSS 24.0 Software (Armonk, NY, USA). The Shapiro–Wilk test was used to determine the normality of the distribution of variables. A two-way repeated measures analysis of variance was used to test the effects of changes in load levels (high and low) and the presence of pain-related or control sounds. Bonferroni correction was applied to handle multiple comparisons. The statistical significance level was equal to 0.05. The differences between SF-MPQ scores in the validation study and the main study were assessed using an unpaired t-test. One participant did not fill out a NASA-TLX questionnaire for a single task; therefore, the results for the NASA-TLX were calculated for 11 participants (five females and six males).

## 4. Results

The results regarding the oculometrics and performance are outlined in Table 2. All variables were normally distributed according to the Shapiro–Wilk tests.

### 4.1. Saccades

The saccade peak velocity decreased as the load level of the cognitive task increased ((1,11) = 6.6, *p* = 0.026). The saccade frequency also decreased significantly with an increased load level ((1,11) = 11.3, *p* = 0.006). No significant effect of pain-related sounds was found for any of the saccadic metrics, nor was there an interaction between the load levels and the induction of pain-related feelings.

### 4.2. Fixations

The fixation duration significantly increased in response to an increasing load level ((1,11) = 19.7, *p* = 0.001). There was no effect of pain induction, nor an interaction between load levels and pain induction related to the fixation duration. The fixation frequency decreased as the load level increased ((1,11) = 25.1, *p* < 0.001). No effect of the induction of pain-related feelings was observed for the fixation frequency, nor was there an interaction between the load levels and the induction of pain-related feelings.

### 4.3. Blinks

The duration and frequency of the blinks did not change significantly with the cognitive load and the induction of pain-related feelings. No interaction was found between the levels of the task load or the induction of pain-related feelings.

### 4.4. Pupillary Responses

The pupil dilation range significantly increased with the load level ((1,11) = 40.4, *p* < 0.001). A tendency for the pupil dilation range to increase in response to the induction of pain-related feelings was observed ((1,11) = 4.3, *p* = 0.063). A significant interaction between the induction of pain-related feelings and the cognitive load was found in the pupil dilation range ((1,11) = 7.4, *p* = 0.020), wherein the increase in the pupil dilation range in response to the presence of pain-related sounds was only observed at the low cognitive load (*p* = 0.007).

### 4.5. Overall Performance

A significant decrease in the overall performance was found with an increased cognitive load ((1,11) = 106.8, *p* < 0.001). There was no effect of pain-related sounds, nor an interaction between the two factors when the overall performance was considered.

### 4.6. Subjective Ratings

As outlined in Table 3, no significant differences in pain perception were observed between the tasks with different cognitive loads or within the same cognitive load level in relation to the SF-MPQ scores. The level of pain perceived from the pain-related sounds acquired in the validation study was significantly higher than that found in the main study for the LP (*t*(24) = 8.8, *p* < 0.001) and HP (*t*(24) = 10.1, *p* < 0.001). The workload of the tasks, as quantified by the total NASA-TLX score, increased significantly with the cognitive load level ((1,10) = 37.2, *p* < 0.001). The pain-related sounds did not affect the workload; no interaction between the two factors was found in the NASA-TLX scores. Figure 2 shows the mean ratings of the six workload subscales of the NASA-TLX across participants. Mental and temporal demands were the dominant subscales in all four tasks.

## 5. Discussion

To our knowledge, this is the first study to investigate the associations between pain-related feelings and cognitive load using oculometrics. In light of the small sample size of this study, the findings need to be interpreted with caution until larger studies are done to substantiate the findings.

According to the oculometrics, the presence of pain-related sounds influenced the pupillary responses, as reflected markedly in the pupil dilation range. The subjective scores from the validation study support the induction of pain-related feelings. However, no significant difference in the self-perception of pain-related feelings was found between the tasks with the control sounds and the tasks with the pain-related sounds in the main study, which involved performing the cognitive task. Although uncontrolled factors might have led to the reduced affective pain scores in the main study compared with the validation study, the effect of performing the cognitive tasks cannot be simply ruled out due to the observation of such a significant reduction. The cognitive load levels were reflected in the saccade peak velocity, saccade frequency, fixation duration, fixation frequency, and pupil dilation range, in line with previous findings [32].

According to a previous study, shorter fixations and less frequent saccades and fixations are associated with nociceptive pain stimulation and are interpreted as limited visual exploratory behavior in response to pain [6]. Similar effects on saccades and fixations in response to cognitive load variation were found in the current study. The pain-related feelings were only reflected in the pupil dilation range during periods of low cognitive load. This may suggest that the participants were distracted from the pain-related sounds by the cognitive load, which is also supported by the non-significant differences in the SF-MPQ and the NASA-TLX scores between the tasks with the control sounds and pain-related sounds. The ocular reflection of pain-related sounds that only occurred in the low cognitive load conditions might be explained by the limited attentional resources to be directed to the pain-related sounds compared with in the tasks with a high cognitive load.

Pupil diameter has previously been reported to be sensitive to stress in response to auditory stimuli [49]. Regardless of the type of pain, affective or sensory, pupillary responses have been shown to be sensitive to pain perception [50,51,52,53,54,55,56,57]. In connection with these findings, the results may suggest the viability of the measurement of pain perception using pupillary responses under conditions of low cognitive load. Video-based eye tracking was used here as a noninvasive modality to study pain-related feelings. The device and the task imposed negligible physical demands on the participants, as reflected in the NASA-TLX scores (Figure 2). Attentional bias to pain-related stimuli can be captured using eye tracking and visual stimulation [6,34,58,59]. In the present study, the utilization of auditory stimuli was proposed to avoid interference in oculometrics that may occur when using visual stimuli to induce pain.

Pain may impair attention and be reflected by a compromised performance in cognitive tasks, e.g., the n-back task [35]. Such an impairment was not observed in the current study, as the overall performance did not decrease significantly in response to the pain-related sounds. This may underline the task-specificity of attention allocation. This could also be due to the attentional differences between physical pain, e.g., in response to noxious stimuli, and pain-related feelings, in that physical pain is rather easy to localize [60] and is considered to cause a certain level of damage compared with pain-related feelings, which might be perceived as a potential threat [61].

The mechanism by which pain memory and pain-related feelings influence pain perception has rarely been investigated. There is, however, evidence to support the occurrence of neural transmission of the affective pain in the brain [13].

The control sounds used in this study were emotionally neutral. It is, however, interesting to investigate how embedding sounds with positive valence [62] or sounds with analgesic effects [63] into a cognitively demanding task may counteract pain perception. In addition, validated inventories for pain-related sounds are lacking. Finding reliable sounds and their characteristics in terms of duration and affective dimensions to induce pain-related feelings could thus be a topic for future studies.

The use of cognitive tasks to distract from pain-related stimuli is could be investigated in individuals suffering from chronic neuropathic pain and related psychiatric comorbidities. According to the sensitivity and the specificity of the oculometrics, participants can be recruited in a biofeedback framework to enhance the rehabilitation or treatment procedure for pain management [64]. Previous studies have reported the sensitivity of oculometrics to changes in cognitive load in both young and elderly individuals [32]. Future studies may elaborate our understanding of age-related and sex-related differences in subjective and objective measures with a larger sample size of different age groups.

## 6. Conclusions

In summary, the non-significant difference in affective pain scores during low- and high-load cognitive tasks including pain-related sounds does not support the hypothesis that the perception of pain-related feelings could be decreased by exposure to an increased cognitive load. The significant decrease in the perceived level of pain-related feelings when performing the cognitive tasks compared with merely listening to sounds may, however, partly support the potential of cognitive processing to decrease the induction of pain-related feelings. Among the oculometrics, the pupillary responses appeared to be more informative concerning the pain-related sounds.

## Figures and Tables

**Figure 1 behavsci-10-00092-f001:**
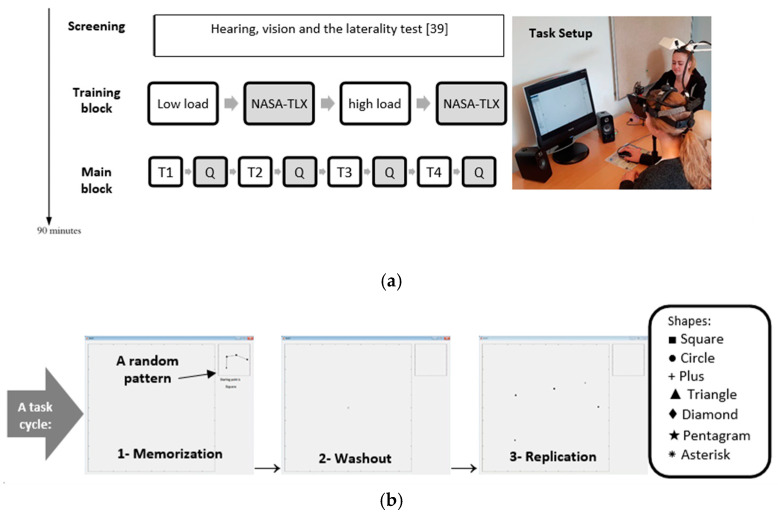
(**a**) Timeline of the experimental procedure including the questionnaires (Q; National Aeronautics and Space Administration Task Load Index (NASA-TLX) and the Short-form McGill Pain Questionnaire (SF-MPQ)). T1, T2, T3, and T4 are 5-min tasks with low- and high-load levels, once with the control sounds and once with the pain-related sounds. The order of the tasks was randomized across participants. (**b**) The three sections of a task cycle (Section 2.4) are depicted, beginning with memorization of a random pattern of connected points in different shapes, a washout period with fixation on a central point, and replication of the memorized pattern using mouse clicks on the points in a correct order. The shapes with the names used in the patterns are also indicated here.

**Figure 2 behavsci-10-00092-f002:**
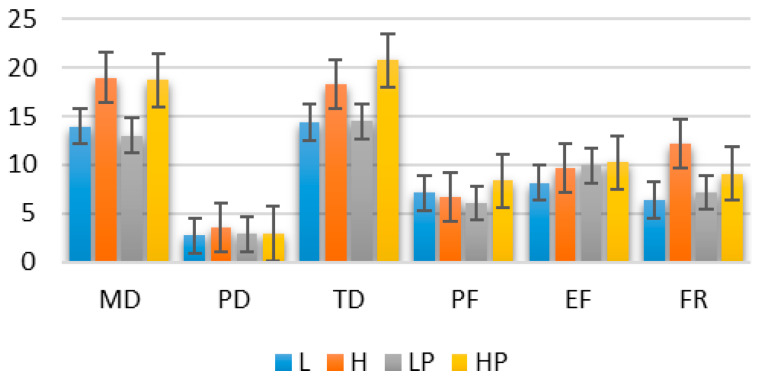
Mean weighted NASA-TLX scores (Y-axis) in the workload subscales across all participants in the four tasks (L, LP, H, and HP). The NASA-TLX subscales are as follows: Mental Demand (MD), Physical Demand (PD), Temporal Demand (TD), Performance (PF), Effort (EF), and Frustration (FR). Error bars denote SDs.

**Table 1 behavsci-10-00092-t001:** List of the selected pain-related and control sounds.

Control Sounds	Pain-Related Sounds
Barred owl	Dental scaler
Heart beats	Dental turbine
Vivid human voice in crowd	Horn
Babbling brook water streaming	Sawing machine
Waterfall	Medical device alarm
Air	Civil defense siren

**Table 2 behavsci-10-00092-t002:** Outcome variables (mean ± SD) measured in four tasks.

Variable	L	LP	H	HP
*Oculometrics*				
**SPV (°/s)**	119 ± 9	116 ± 10	113 ± 6	114 ± 9
**SF (Hz)**	1.35 ± 0.21	1.43 ± 0.19	1.26 ± 0.18	1.25 ± 0.22
**FD (s)**	0.47 ± 0.10	0.48 ± 0.10	0.51 ± 0.10	0.52 ± 0.10
**FF (Hz)**	1.8 ± 0.3	1.8 ± 0.3	1.7 ± 0.3	1.7 ± 0.3
**PDR (mm)**	0.31 ± 0.11	0.35 ± 0.13	0.38 ± 0.13	0.39 ± 0.14
**BD (s)**	0.19 ± 0.03	0.18 ± 0.03	0.18 ± 0.03	0.18 ± 0.03
**BF (Hz)**	0.20 ± 0.18	0.19 ± 0.15	0.19 ± 0.16	0.18 ± 0.15
*Performance*				
**OP**	0.42 ± 0.10	0.43 ± 0.11	0.27 ± 0.10	0.29 ± 0.14

Low cognitive load without pain-related sounds (L), low cognitive load with pain-related sounds (LP), high cognitive load without pain-related sounds (H), high cognitive load with pain-related sounds (HP), saccade peak velocity (SPV), saccade frequency (SF), fixation duration (FD), fixation frequency (FF), pupillary dilation range (PDR), saccade amplitude (SA), blink duration (BD), blink frequency (BF), and overall performance (OP).

**Table 3 behavsci-10-00092-t003:** The means ± SDs of the scores from the short-form McGill pain questionnaire (SF-MPQ).

	Main Study				Validation Study	
	L	LP	H	HP	PR	C
Pain scores	0.6 ± 0.6	0.9 ± 1.1	1.1 ± 0.7	0.8 ± 1.0	3.5 ± 0.1	1.8 ± 0.5

Low cognitive load without pain-related sounds (L), low cognitive load with pain-related sounds (LP), high cognitive load without pain-related sounds (H), and high cognitive load with pain-related sounds (HP), pain-related sounds (PR), and control sounds (C).

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
