# Peer review of "Eye Movements in Response to Pain-Related Feelings in the Presence of Low and High Cognitive Loads"

_behavsci, 2020, doi:10.3390/bs10050092_

Round 1

Reviewer 1 Report

This is an interesting that investigates the relationship between cognitive load and pain-related feelings. The authors novelly used eye movement as a measurement for cognitive load and pain-related feelings. A total of 12 participants had joined the study. The result only showed a negative association between cognitive load and pain-related feelings.

Generally, the logic behind the manuscript is easy to followed. There are some comments for the authors.

  1. The authors may consider revising their title and abstract. The title reads like the authors will examine the interaction among 3 factors including cognitive load, pain feelings, and eye movements. If I understand correctly, eye movement, the innovative part, is a measure of cognitive load and pain-related feelings.
  2. The introduction part is lengthy in the current version.
  3. I was confused by the repeated twp-way ANOVA. An interaction plot might be helpful to illustrate the key finding. Moreover, the authors may considered  a simpler test such as wilcoxon sign rank test first to examine the association between coginitive loading and pain feeling.
  4. If the relationship does not exist, it's still very valuable to focus on the methodology.

Author Response

Point-by-point response letter to the reviewers:

Comments and Suggestions for Authors #1

This is an interesting that investigates the relationship between cognitive load and pain-related feelings. The authors novelly used eye movement as a measurement for cognitive load and pain-related feelings. A total of 12 participants had joined the study. The result only showed a negative association between cognitive load and pain-related feelings. Generally, the logic behind the manuscript is easy to followed. There are some comments for the authors.

Response: We thank the reviewer’s positive view. We have addressed below the points raised by the reviewer to enhance the quality and clarity.

  1. The authors may consider revising their title and abstract. The title reads like the authors will examine the interaction among 3 factors including cognitive load, pain feelings, and eye movements. If I understand correctly, eye movement, the innovative part, is a measure of cognitive load and pain-related feelings.

 Response: We understand the concern of the reviewer about the title and abstract. Both tile, and abstract, have now been modified in the revised version of the article to reflect on the novelty and findings from the study.

The new title: “Eye Movements in Response to Pain-Related Feelings in the Presence of Low and High Cognitive Loads”

The new abstract: 

The affective dimension of pain contributes to pain perception. Cognitive load may influence pain-related feelings. Eye tracking has proven useful for detecting cognitive load effects objectively by using relevant eye movement characteristics. In this study, we investigated whether eye movement characteristics differ in response to pain-related feelings in the presence of low and high cognitive loads. A set of validated, control, and pain-related sounds were applied to provoke pain-related feelings. Twelve healthy young participants (six females) performed a cognitive task at two load levels, once with the control and once with pain-related sounds in a randomized order. During the tasks, eye movements and task performance were recorded. Afterwards, the participants were asked to fill out questionnaires on their pain perception in response to the applied cognitive loads. Our findings indicate that an increased cognitive load was associated with a decreased saccade peak velocity, saccade frequency, and fixation frequency, as well as an increased fixation duration and pupil dilation range. Among the oculometric data, pain-related feelings were reflected only in the pupillary responses to a low cognitive load. The performance and perceived cognitive load decreased and increased, respectively, with the task load level and were not influenced by the pain-related sounds. Pain-related feelings were lower when performing the task compared with when no task was being performed in an independent group of participants. This might be due to the cognitive engagement during the task. This study demonstrated that cognitive processing could moderate the feelings associated with pain perception.

  1. The introduction part is lengthy in the current version.

Response: Yes. The introduction has now been revised to a shorter length. We tried to revise in such a way that the introduction remains meaningful for the readers. Please see the revised version of the article.

  1. I was confused by the repeated twp-way ANOVA. An interaction plot might be helpful to illustrate the key finding. Moreover, the authors may considered  a simpler test such as wilcoxon sign rank test first to examine the association between coginitive loading and pain feeling.

 Response: We first tested our data set for normality to determine presentation of our data, and statistical tests. Shapiro–Wilk test was used to determine the normality and because out data set showed normal distribution, we presented data with average (mean) and the variations with standard deviation (SD). Because data were normally distributed, we could use parametric tests for comparison and because we had more than two column of data for comparison, we needed an ANOVA. Factors were cognitive load (with two levels of high and low), and the presence or absence of pain-related sounds. Time was the repeated factor.

We did not use multiple comparisons separately to reduce the type 2 error. We did not use the Wilcoxon sign rank test, because data were normally distributed and we aimed at comparison of more than two columns of data.

  1. If the relationship does not exist, it's still very valuable to focus on the methodology.

 Response: That is right. We believe that this study is the first within affective computing of pain and hence can be considered a pilot or preliminary.

Reviewer 2 Report

The paper was very hard to follow as it had space between characters of many words for example tis- sue, psychophys- iological and many other words.

The sample size is very small. How the results can be generalised and what are the limitations of the study.

How lines 346-347 can be concluded from the paper.

What is the significance of the paper in comparison to the existing research especially there were many non statistically significant relationships in the results.

Author Response

Comments and Suggestions for Authors #2

The paper was very hard to follow as it had space between characters of many words for example tis- sue, psychophys- iological and many other words.

Response: That is right. We are not aware how this has occurred. There must have been a technical error, and we apologize for that. We have now proofread the text thoroughly and fixed those errors.

The sample size is very small. How the results can be generalised and what are the limitations of the study.

Response: We believe that this study is the first within affective computing of pain and hence can be considered as a pilot or preliminary. We have added this into start of discussion to emphasize on the small sample size and that larger studies are required to substantiate findings from the present study.

How lines 346-347 can be concluded from the paper.

Response: The lines stated, “The current results may support the importance of attention, which can be succeeded by memory formation in pain perception”. What we meant is that high cognitive load that took away the attention, led to scoring less in response to pain-related sounds. However, we can understand the concern of the reviewer that with the techniques applied in this study this cannot be speculated. Hence, we deleted the statement to avoid confusion.

What is the significance of the paper in comparison to the existing research especially there were many non-statistically significant relationships in the results.

Response: We believe that this study is the first within affective computing of pain and hence can be considered a pilot or preliminary. Small sample size and other limitations that have been mentioned in the discussion can stimulate curiosity for further research in larger cohorts. Among the oculometrics, the pupillary responses appeared to be more informative concerning the pain-related sounds. However, other characteristics may show up significant by application of methods in chronic pain patients or other experimental pain models that can consider other aspects of pain perception. Combination of somatic and affective sensations might have stronger influence on oculometrics, in particular in healthy volunteers that have no memory of long-term pain.

Round 2

Reviewer 1 Report

This is the revised version of  manuscript (#behavsci-782357). The authors had carefully addressed all issues and made their manuscript more concise. This study provided an clue for pain control for clinical use.

Reviewer 2 Report

Authors answered all my questions.